# How Accurate Is the Diagnosis of “Chronic Obstructive Pulmonary Disease” in Patients Hospitalized with an Acute Exacerbation?

**DOI:** 10.3390/medicina59030632

**Published:** 2023-03-22

**Authors:** Fares Darawshy, Ayman Abu Rmeileh, Rottem Kuint, Polina Goychmann-Cohen, Zvi G. Fridlender, Neville Berkman

**Affiliations:** 1Faculty of Medicine, Hebrew University of Jerusalem, Jerusalem 9112102, Israel; 2The Institute of Pulmonary Medicine, Hadassah Medical Center, Jerusalem 91000, Israel

**Keywords:** COPD, chronic obstructive pulmonary disease, spirometry, GOLD

## Abstract

*Rationale*: COPD diagnosis requires relevant symptoms and an FEV1/FVC ratio of <0.7 post-bronchodilator on spirometry. Patients are frequently labeled as COPD based on clinical presentation and admitted to the hospital with this diagnosis even though spirometry is either not available or has never been performed. The aim of this study was to evaluate the accuracy of COPD diagnosis based on post-bronchodilator spirometry, following hospital admission for COPD exacerbation. *Methods*: This is a retrospective study with a cross-sectional analysis of a subgroup of patients. Demographic and clinical data and pre-admission spirometry were collected from electronic records of patients hospitalized with a primary diagnosis of COPD. Patients without available spirometry were contacted for a pulmonary consultation and spirometry. Three groups were compared: patients with a confirmed COPD diagnosis (FEV1/FVC < 0.7), without COPD (FEV1/FVC > 0.7), and those who have never performed spirometry. *Results*: A total of 1138 patients with a recorded diagnosis of COPD were identified of which 233 patients were included in the analysis. Only 44.6% of patients had confirmed COPD according to GOLD criteria. In total, 32.6% of the patients had never undergone spirometry but were treated as COPD, and 22.7% had performed spirometry without evidence of COPD. Recurrent admission due to COPD was a strong predictor of a confirmed COPD diagnosis. *Conclusions*: Among the patients admitted to the hospital with a COPD diagnosis, a high proportion were not confirmed by the current GOLD report or had never performed spirometry. Stricter implementation of the diagnostic criteria of COPD in admitted patients is necessary to improve diagnosis and the treatment outcomes in these patients.

## 1. Introduction

The estimated prevalence of chronic obstructive pulmonary disease (COPD) may vary according to country, ranging between 8.6–13.6% in China [1] and 6.44–8.07% in the United States [2]. The global prevalence of COPD in 2016 was estimated to be 251 million cases globally [3], and the number of COPD associated deaths has doubled over the last two decades. COPD is the fourth leading cause of mortality worldwide [3,4].

According to the Global Initiative for Chronic Obstructive Lung Disease (GOLD) report [5], the diagnosis of COPD requires relevant symptoms and the performance of spirometry, with documentation of the forced expiratory volume in one second and a forced vital capacity (FEV1/FVC) ratio less than 0.70 post-bronchodilators. However, in practice, especially among family physicians, a diagnosis of COPD is often made in patients with a history of smoking and the presence of typical symptoms (dyspnea, chronic cough, sputum production) and physical findings (expiratory wheezing), without the performance of spirometry [6]. This is inaccurate and may lead to under or over-diagnosis of COPD and inappropriate treatment. In addition, clinical diagnosis and the International Classification of Diseases (ICD) codes for COPD are often used in hospitalized patients without validation of the diagnosis by spirometry. This is particularly problematic in COPD, which can easily be misdiagnosed in patients with heart failure, asthma, bronchitis, vocal cord dysfunction, sleep apnea, and morbid obesity [7,8,9,10,11,12,13,14,15].

The burden of COPD related care in emergency departments (ED) and hospitals remains significant. However, rates of hospitalization may vary among different populations. In a study that evaluated the frequency of COPD related ED visits and hospital admissions in North Carolina, the annual rate of ED visit was 13.8 per 1000 person-years. Among patients with COPD, 51% were admitted to the hospital from the index ED visit [16]. In a Danish cohort, the rate of the first hospitalization of COPD was 231 per 100,000 person-years, with more deaths occurring within 180 days in COPD patients compared to a matched control cohort (16% vs. 2.4%) [17]. In another study from England, the mean annual COPD admission rates ranged from 124.7 to 646.5 per 100,000 population [18]. These studies concluded that COPD is a major public health problem leading to significant hospitalization rates and substantial mortality.

Previous studies have investigated both the use of spirometry and the accuracy of COPD diagnosis in community and hospital care. One large study [19] evaluated 701 subjects to assess COPD diagnosis accuracy in primary care and reported a 13% overdiagnosis and a 59% underdiagnosis of COPD. General physicians were able to correctly exclude patients who did not have COPD but were less accurate in diagnosing COPD patients [19]. In another study that evaluated patients with COPD in primary care, a spirometry report was available in only 58% of patients, with a COPD diagnosis confirmed in only 75% of them. These studies concluded that COPD is often misdiagnosed in primary care. 

The accuracy of COPD diagnosis in hospitalized patients varies between studies. The performance rate of spirometry in hospitalized patients with COPD as a primary diagnosis (and who received treatment as COPD) ranges from 35% to 69.2% [13,14,15]. In a European and UK national COPD audit, a diagnosis of COPD was confirmed in 46% to 51% of cases [20,21].

In the present study, we evaluate the accuracy of a diagnosis of COPD based on post-bronchodilator spirometry in patients discharged with this diagnosis at a single tertiary academic medical center (Hadassah Medical Center, Jerusalem, Israel). 

## 2. Methods

### 2.1. Study Design

This is a retrospective single-center study with a cross-sectional analysis of a subgroup of patients, performed at Hadassah Medical Center (Jerusalem, Israel). Electronic records of hospitalized patients for the years 2015–2018 were reviewed. 

An official local ethics committee (Institutional Review Board: Hadassah Medical Center Helsinki committee) approved the study protocol with the reference number 0042-19-HMO. A waiver was applied for all retrospective data, which was obtained from electronic health records. Written informed consent was obtained from patients who attended the clinic.

We identified the charts’ ICD Ninth Revision (ICD-9) coding, searching for COPD as the primary diagnosis during hospital admission. We excluded any re-hospitalizations for COPD in the study period from the analysis. 

### 2.2. Retrospective Data Collection

We reviewed patients’ records for the presence of spirometric measurements and evidence of airway obstruction (FEV1/FVC < 0.7 post-bronchodilators) at any time prior to hospitalization. The largest values for FEV1 and FVC, in the case of multiple values prior to hospitalization, were used (according to the American Thoracic Society/European Respiratory Society standards [22,23]). The prediction equations used during the study period were the European Community for Steel and Coal (ECSC).

We collected data from electronic medical records, including demographic, historical, clinical, and radiological data. Demographic data included age, sex, and race. Historical data included documentation of a consultation with a pulmonary/respiratory specialist prior to hospitalization, current use of inhalers, and past medical history. Historical data also included smoking status, which was defined as follows: A current smoker was defined as a smoker at the time of admission, a past smoker defined as someone who had smoked at least one pack-year and had stopped smoking up to 1 month prior to hospitalization. A never smoker was defined as someone with no significant personal history of smoking (less than a total of one pack-year). 

Clinical data during hospitalization included laboratory data (C-reactive protein, white blood cells count, and blood gas analysis upon admission), treatment given during or after hospitalization (bronchodilator therapy, antibiotics, diuretics, and systemic glucocorticoids), the need for respiratory support (both invasive and non-invasive ventilation) and a pulmonary specialist consultation during and after hospitalization. Radiological data included computed tomography (CT) scan results before the admission. 

### 2.3. Follow Up Visit

Patients diagnosed with COPD during hospital admission in whom spirometry was not available, were contacted by telephone after discharge either by a pulmonary physician, pulmonology nurses, or pulmonary function lab technician. Patients were requested to respond to the following questions: Are you aware that you have been diagnosed with COPD during hospitalizations? What is your smoking status? Have you consulted a pulmonary physician in the past? Do you receive any inhaler therapy, and if so, which? Have you performed a CT scan of the chest in the past? Have you had a prior hospital admission for COPD or any other reason in the last three years? Have you performed spirometry in the past? If the patient had not performed spirometry, he/she was asked to attend a pulmonary clinic visit, which included a pulmonary specialist consultation and pre/post-bronchodilator spirometry. 

### 2.4. Comparison of Patients Groups Based on Spirometry

We divided the patients into three groups based on spirometry, namely, patients with confirmed COPD diagnosis (FEV1/FVC < 0.7), patients without COPD (FEV1/FVC ≥ 0.7), and patients who never had spirometry. The last group included patients who definitely never performed spirometry, after reviewing their medical files, contacting attempts and specifically questioning them in this regard. Patients with missing information, or those that could not be contacted, were excluded from the analysis.

To reduce the bias of age, sex and height, a further analysis was carried out (when the height variable was available, in 89 patients overall) using the Global Lung Initiative 2012 to calculate the Z-score and the lower limit of normal (LLN) for each spirometry index measured (FEV1, FVC and FEV1/FVC), and these were compared between the patient groups. 

### 2.5. Data Analysis

Descriptive statistics were presented as the means and standard deviation (Mean ± SD) to describe the sample’s demographic and clinical characteristics. Comparisons of the continuous variables, such as patient characteristics and spirometry findings, were analyzed via Kruskal–Wallis (KW) tests while Chi-square tests were used for the categorical variables. Mann–Whitney tests with a Bonferroni correction were used as post hoc tests for the continuous variables. To assess the most important variables associated with a confirmed COPD diagnosis, we performed a univariate analysis followed by a logistic regression model while comparing two distinct groups, namely, patients with confirmed COPD diagnosis, and the two groups of patients with no COPD and no spirometry measures. The factors inserted in the model were the variables that were significantly associated with having COPD in the univariate analysis. *p*-values < 0.05 were considered significant. The *p*-value of the Bonferroni correction was 0.0045 for the KW significant results. 

## 3. Results

After excluding readmissions, a total of 1138 COPD patients were identified (Figure 1). Of these, 827 were excluded due to COPD not being the primary diagnosis of admission, and another 78 were excluded due to missing information that could not be obtained. In total, 233 patients were included in the final analysis after the follow up visit. Of these, 157 had available spirometry, 104 (44.6%) in the confirmed COPD group, and 53 (22.7%) in the group without COPD (11 of the patients included in those two groups, in whom spirometry was performed, presented to the clinic). Overall, 66.2% of those who had spirometry were found to be obstructive. The remaining 76 (32.6%) patients were included in the no spirometry group. Out of these, 52 responders refused to present to the clinic, and 24 were deceased but had valid information in their file confirming the absence of spirometry. 

### 3.1. Patients’ Characteristics 

Patients’ characteristics are described in Table 1. The majority of our cohort were male patients in advanced age. The patients presented with a range of comorbidities, including hypertension, heart disease, and asthma, with hypertension being the most common. Smoking was also highly prevalent, with a majority of patients being current or former smokers. In over half of our cohort, the diagnosis of COPD was established by a pulmonary specialist, and chest CT scans were performed, with emphysema being present in fewer than a third of the patients. Inhaler medications were used in varying degrees, with triple therapy being the most common treatment among those who were prescribed inhalers. 

### 3.2. Differences between the Patient Groups

Full results of the comparison between the patient groups are given in Table 1. The misdiagnosis of COPD was similar in men and women. On average, the patients without COPD were younger than those with confirmed COPD and those who never had spirometry (Mean = 65.5 vs. 71.5 and 71.0, *p* < 0.01 and =0.01, respectively). Of the Jewish patients, a higher percentage had a confirmed COPD diagnosis (75%) in comparison to Arab patients (25%); however, spirometry was less often requested for Arab patients than their Jewish counterparts (24.8% vs. 47.4%, *p* < 0.01). Nonsmokers (7.9%) and past smokers (21.1%) were less likely to perform a spirometry in comparison to current smokers (71%) (*p* < 0.01). The majority of patients diagnosed by a pulmonary physician were indeed confirmed as having COPD (82.7%), but most of those diagnosed by other physicians (51.3%) never had spirometry performed. Patients with confirmed COPD (42.3%) tended to have more emphysema on a chest CT scan in comparison to the patients without COPD (20.8%), and those without spirometry (*p* = 0.056, and *p* = 0.03, respectively), and had higher rates of readmissions compared to the other groups (67.3% vs. 47.2% and 53.9%) (*p* = 0.014 and *p* = 0.06, respectively). In addition, the reason for recurrent admission after the index admission was more often due to COPD exacerbation in patients with confirmed COPD, compared to those without COPD (27.9% vs. 5.6%, *p* = 0.0018), but not significantly more than those without spirometry (*p* = 0.13).

Statistically significant differences were found in the absolute FEV1 and FVC values and Z-score for both indices, being lower in the confirmed COPD group. LLN calculation using the GLI-2012 was available in 62 patients in the confirmed COPD group, and in 27 patients in the no COPD group. In total, 61 patients had FEV1 below the lower limit of normal (LLN) in the confirmed COPD group, and only 22 in the group without COPD. The FEV1/FVC ratio was also significantly lower in the confirmed COPD group; 49 patients (out of 62) had FEV1/FVC below the LLN, and 13 were with FEV1/FVC > LLN. In contrast, none of the patients in the groups without COPD had FEV1/FVC below the LLN. 

Statistically significant differences were also found in the baseline treatment; the majority of patients with confirmed COPD were on triple inhaler therapy while triple therapy was less frequent in patients without COPD and in those who never had spirometry (42.3% vs. 18.9% and 1.3%; *p* < 0.01 and *p* < 0.001, respectively). A higher number of patients, in whom spirometry was never performed, did not receive any inhaler therapy compared to other groups (73.3% vs. 15.4% and 45.3%; *p* < 0.01 and *p* = 0.04, respectively). 

### 3.3. Admission Course

There were no differences between the groups regarding the treatment given during admission (Table 2). The majority of patients were treated with antibiotics, systemic steroids, and short acting bronchodilators (using nebulizers), regardless of having known spirometry results. In addition, COPD diagnosis remained on patients’ charts after discharge, regardless of the spirometry results prior to their admission. 

Patients with confirmed COPD were more likely to have had a pulmonary physician consultation during their admission than those without spirometry (25% vs. 9.2%, *p* = 0.006), but not significantly more than those without COPD (*p* = 0.2). Confirmed COPD patients had more consults following admissions compared to other groups (*p* = 0.002 and *p* < 0.01). More patients required respiratory support (both invasive and non-invasive mechanical ventilation) in the confirmed COPD (26.0%) group in comparison to other groups, although this did not reach statistical significance (Table 2). 

### 3.4. Laboratory Results

Patients with confirmed COPD had higher white blood cell (WBC) counts at presentation (Mean = 12.34 WBC/µL), in comparison to the patients without COPD (Mean = 10.6, padj < 0.01). Blood gas analysis at admission showed higher PCO2 (Mean = 53 mmHg) in confirmed cases, compared to patients without COPD (Mean = 44.5 mmHg, padj < 0.01). There was no difference between the groups concerning the C-reactive protein level. 

### 3.5. Factors Associated with Having a Confirmed COPD Diagnosis (Table 3)

The logistic regression results are shown in Table 3. Only two variables were significantly associated with having confirmed COPD. These were the occurrence of recurrent admissions due to COPD (OR = 7.1, CI [1.25, 40.51], *p* = 0.03), and the presence of a formal pulmonologist consultation following admission (OR = 14.95, CI [2.41, 88.22], *p* < 0.001). 

**Table 3 medicina-59-00632-t003:** Multivariate analysis of factors associated with having a confirmed COPD diagnosis.

	OR	CI		*p*-Value
		Lower	Upper	
Age	1.03	0.95	1.12	0.49
Origin	1.66	0.34	8.18	0.53
Smoking	0.93	0.17	5.06	0.93
Past Smoker	1.5	0.04	52	0.82
Chronic Oxygen Therapy	1.13	0.23	5.63	0.88
CT before admission	0.37	0.08	1.69	0.20
Emphysema on CT before admission	2.44	0.6	9.99	0.21
Pulmonologist follow up before admission	0.25	0.01	4.88	0.36
Inhaler therapy	1.81	0.23	14.18	0.57
WBC	1.02	0.97	1.07	0.52
PCO2	1.05	0.99	1.11	0.08
Pulmonologist consultation during admission	4.78	0.55	41.47	0.16
Pulmonologist consultation following admission	14.59	2.41	88.22	<0.001
Recurrent admission due to COPD	7.1	1.25	40.51	0.03
Number of Hospitalization days	1.05	0.99	1.13	0.13

Multivariate analysis of parameters predicting a true COPD diagnosis. OR = Odds ratio; CI = 95% confidence interval; WBC = White blood cell count; PCO2 = partial pressure of carbon dioxide.

## 4. Discussion

In this study, we found that only 44.6% of patients admitted for acute exacerbations of COPD had confirmed COPD according to the GOLD report criteria and 32.6% of the patients had never undergone spirometry prior to admission. A total of 22.7% of the patients hospitalized and treated as COPD had no evidence of COPD according to GOLD report criteria [5], and were, in fact, misdiagnosed. We found that a history of recurrent exacerbations due to COPD was associated with a COPD diagnosis confirmed by spirometry. Our finding that 66.2% of patients who performed a spirometry were found to have an obstructive result, highlights the importance of confirming COPD diagnosis. 

Our findings relating to the frequency of spirometry performance among patients hospitalized with a diagnosis of COPD are concordant with the findings reported in previous studies [6,13,14,15,19], although the range varies considerably according to the population and study cohort. Wu et al. reported 8% of patients without airflow obstruction and 21% in whom spirometry was not performed [15]. Spero et al. reported that up to a third of patients diagnosed and treated as COPD in the hospital might be inaccurately diagnosed as COPD based on confirmatory spirometry [13]. Differences between findings across studies can likely be attributed to multiple factors, including differences in referral patterns to pulmonary specialists, availability, utilization of spirometry as a tool for COPD diagnosis, and awareness or training among primary care physicians regarding COPD diagnosis. Emphasizing the above is the fact that 51.3% of the patients who never had spirometry were diagnosed by physicians other than a pulmonary specialist. In addition, the majority of our cohort had multiple comorbidities that can mimic COPD, such as heart disease, asthma, and chronic kidney disease. 

In our cohort, 79% of patients in the COPD group with height data had a FEV1/FVC below the LLN, suggesting that 21% may not have COPD according to the ATS/ERS criteria using GLI-2012. This is a significant overdiagnosis by GOLD criteria, emphasizing the need to rely more on LLN in COPD diagnosis. However, these findings cannot be extrapolated to all cohorts or populations due to the lack of height data in other patients. None of the patients in the “not-COPD ” group (as defined by a ratio ≥0.70) had a FEV1/FVC ratio below LLN, suggesting that patients in this group are probably not COPD according to both the GOLD and ATS/ERS criteria. Previous studies have demonstrated a low concordance between GOLD and LLN, particularly as age increases, and this aligns with our findings. 

Confirming the diagnosis of COPD by inpatient spirometry performance is currently not recommended. We argue that spirometry performance in patients admitted with suspected acute COPD exacerbation may increase diagnosis accuracy, rule out COPD in patients with other comorbidities, and promote referral for specialist consult. In addition, it may provide an opportunity for adequate treatment to be given to these patients. Recent studies have shown that spirometry performance during hospitalization, in patients admitted for COPD exacerbation, has a sensitivity of 94% and a positive predictive value of 83% for predicting outpatient airflow obstruction. These studies prove that inpatient spirometry is a valid and reproducible method and provides the opportunity to identify patients admitted with suspected acute COPD exacerbation who have no prior spirometric documentation [24,25]. We suggest that the role of spirometry during hospitalization, in patients suspected to have COPD, should be reconsidered in further prospective studies. 

We found that spirometry requests were low amongst Arab patients admitted with a diagnosis of COPD; 47.4% of the Arab patients had not performed spirometry at all compared to only 25.1% without spirometry amongst Jewish patients (percentage as part of the Arab and Jewish populations, respectively). Previous studies have shown disparities in health care between the Jewish and Arab populations in Israel [26], which affects COPD misdiagnosis due to less access to healthcare facilities, including pulmonary specialists and spirometry testing. Presumably, the level of education and sociocultural factors may also contribute to differences between population groups. The importance of improved diagnosis of COPD in the Arab population is especially important given the very high smoking rates, up to 45%, amongst Arab males. Despite this, the ethnic origin was not a predictor of a true COPD in a multivariate analysis.

Approximately 20% of the patients admitted to the hospital with a diagnosis of COPD without diagnostic spirometry (20.8% in those with non-obstructive spirometry, and 17.1% in whom spirometry was not performed) have radiological evidence of emphysema. Based on previous definitions (prior to 2006), these patients would have been considered to have COPD. Current GOLD report criteria do not adequately address this group of patients, and there are no clear guidelines as to how these patients should be classified or treated. The GOLD report considers emphysema as a radiological and pathological finding that may contribute to airflow limitation, rather than a disease, and recommends repeated spirometry performance in these patients. In addition, GOLD recommends interventional treatment in severe cases of emphysema, but without a clear recommendation on bronchodilator therapy. Ongoing research attempts to identify patients with emphysema and early COPD by other means (e.g., combining other spirometry indices and chest imaging) [27,28,29]. These studies highlight that patients with emphysema already have or may develop COPD even in the presence of non-obstructive spirometry. Furthermore, we found that patients with an FEV1/FVC ≥ 0.7 might have low FEV1 < 80%. These patients are diagnosed without COPD according to GOLD criteria. However, they might be diagnosed with COPD according to the COPDGene definition [29], indicating, again, that some patients with non-obstructive spirometry already have or may develop COPD. 

We found several apparent differences between patients with confirmed COPD compared to the other two groups. These patients were more likely to be current or past smokers, have emphysema on a chest CT scan, have been diagnosed by a pulmonary specialist, suffer from recurrent admissions due to COPD exacerbation, have lower spirometry values, and are more likely to be treated with triple inhaler therapy. However, when assessing the factors that are associated with a spirometry confirmed COPD diagnosis, we found that a strong factor was recurrent admission due to COPD exacerbation. These findings may be explained by the fact that severe patients with advanced disease, lower spirometry values and recurrent exacerbations, are more likely to present to healthcare facilities, be examined by a pulmonologist during their admission, be diagnosed with COPD by a pulmonologist according to spirometry, and to receive triple inhaler therapy due to their advanced condition. On the other hand, patients without recurrent exacerbations, who are clinically stable and without prominent emphysema, either do not have COPD or will get less attention from their primary care physician and will not perform spirometry or be sent to a pulmonary specialist consult. Nevertheless, other factors that can affect the treatment choice were not obtained in our study, including blood eosinophils count, the combination of triple therapy, and dyspnea scores (such as the COPD assessment score). 

Although we found significant differences between the three groups of patients regarding spirometry performance and the accuracy of COPD diagnosis, there were no differences between the groups in the treatment given during their admission. Therefore, the treatment of patients suspected to have COPD in our hospital is probably based on clinical presentation and smoking history, rather than a confirmed diagnosis of COPD based on the GOLD criteria [12]. We do not know if the availability of spirometry and the correct diagnosis of our patients would have changed the inpatient management (we assume it would), nor do we know whether having a spirometry-based diagnosis would have impacted upon patient outcomes. This is certainly a subject for future studies. 

Our study has several limitations. Firstly, this is a single center study. Secondly, not all patients admitted with COPD could be contacted or returned for follow up evaluation (due to death, could not be contacted, or refused to do so), which may have introduced a selection bias. Thirdly, the quality aspects of spirometry were not assessed, and we focused mainly on the FEV1/FVC ratio as a criterion to differentiate between the patient groups, although the comparison of the LLN and Z-scores strengthened our results. Additionally, we did not have all height data on all patients to calculate the LLN according to GLI-2012. Fourth, other aspects of COPD diagnosis were not assessed, including measurement of lung volume, gas transfer and hyperinflation. Finally, despite our best efforts to gather information about all patients, those without spirometry did not have a documented test and may have performed one at some point, but we could not obtain it.

## 5. Conclusions

Among patients admitted to the hospital with a COPD diagnosis, a high proportion do not have COPD as defined by current GOLD criteria or have never performed spirometry. In our study, there were significant differences in the prevalence of spirometric testing amongst different ethnic groups. Stricter implementation of diagnostic criteria for diagnosing COPD in both admitted and ambulatory patients will hopefully improve the treatment and outcomes in these patients.

## Figures and Tables

**Figure 1 medicina-59-00632-f001:**
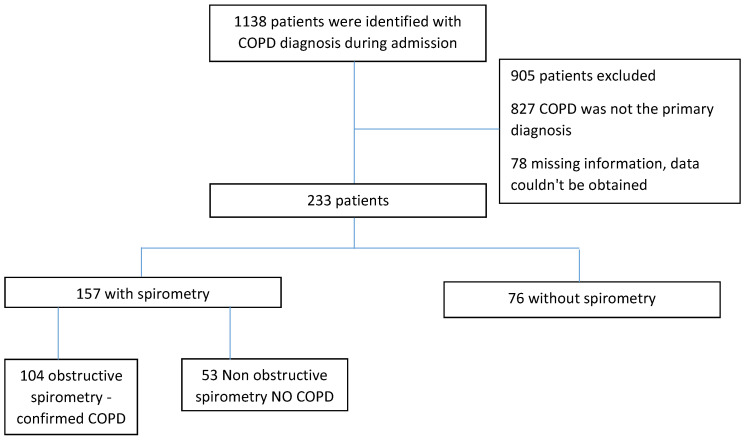
Flowchart showing the distribution of participants and their classification in the different groups and subgroups.

**Table 1 medicina-59-00632-t001:** Comparison between the patient groups.

	N (%) or Mean ± SD				
	Total (N = 233)	Confirmed COPD (N = 104)	No COPD (N = 53)	Never had Spirometry (N = 76)	*p*-Value
Sex					0.58
Male	167 (71.7)	78 (75)	36 (67.9)	53 (69.7)	
Female	66 (28.3)	26 (25)	17 (32.1)	23 (30.3)	
Age (years)	70 ± 11.9	71.5 ± 10.3	65.5 ± 11.9	71 ± 13.0	<0.01
Origin					<0.01
Arab	78 (33.5)	26 (25)	15 (28.3)	37 (48.7)	
Jew	155 (66.5)	78 (75)	38 (71.7)	39 (51.3)	
Past Medical History					
Hypertension	129 (55.4)	54 (51.9)	30 (56.6)	45 (59.2)	0.91
Heart Disease	97 (41.6)	39 (37.5)	22 (41.5)	36 (47.4)	0.41
Asthma	22 (9.4)	10 (9.6)	6 (11.3)	6 (7.9)	0.94
CKD #	30 (12.9)	15 (14.4)	4 (7.5)	11 (14.5)	0.42
Sleep Apnea	19 (8.2)	11 (10.6)	4 (7.5)	4 (5.3)	0.43
Smoking					<0.01
Current	132 (56.7)	52 (50)	26 (49.1)	54 (71)	
Never	16 (6.9)	4 (3.8)	6 (11.3)	6 (7.9)	
Past	85 (36.5)	48 (46.2)	21 (39.6)	16 (21.1)	
Emphysema (by CT)	68 (29.2)	44 (42.3)	11 (20.8)	13 (17.1)	0.01
Length of Stay (Days)	4.3 ± 5.3	3 ± 3.0	4 ± 5.4	7.4 ± 7.4	0.04
Diagnosis Made by					< 0.01
Unknown	38 (16.3)	4 (3.8)	6 (11.4)	28 (36.8)	
Pulmonologist	126 (54.1)	86 (82.7)	31 (58.4)	9 (11.9)	
Other *	69 (29.6)	14 (13.5)	16 (30.2)	39 (51.3)	
Recurrent Admission	136	70	25	41	0.03
	(58.4)	(67.3)	(47.2)	(53.9)	
Recurrent Admission due to COPD	41 (17.6)	29 (27.9)	3 (5.6)	9 (11.8)	<0.01
FEV1 (L)	2.2 ± 0.7	1.1 ± 0.4	1.8 ± 0.6	-	<0.001
FEV1, (% of predicted)	48.6 ± 12.1	45.1 ± 16.9	49.3 ± 18.3	-	0.12
FEV1 < LLN #	79 (34)	61 (58.6)	22 (41.5)	-	<0.001
FEV1 (Z-score) #	−2.85 (−3.7; −2.15)	−2.99 (−4.05; −2.6)	−2.15 (−2.85; −1.29)	-	<0.001
FVC (L)	2.2 ± 0.7	2.1 ± 0.7	2.4 ± 0.8	-	<0.001
FVC (% of predicted)	69.6 ± 21.3	66.5 ± 19.9	75.1 ± 22.4	-	<0.001
FVC < LLN #	61 (65.5%)	42 (40.4)	19 (35.9)	-	0.36
FVC (Z-score) #	−2.31 (−2.9; −1.36)	−2.34 (−2.8; −1.21)	−2.08 (−3; −2.08)	-	<0.001
FEV1/FVC	0.6 ± 0.1	0.5 ± 0.1	0.8 ± 0.1	-	<0.001
FEV1/FVC < LLN #	49 (21)	49 (47.1)	0 (0)	-	<0.001
FEV1/FVC (Z-score) #	−2 (−3.5; −0.58)	−2.72 (−4.05; −1.99	−1.3 (−0.83; 0.47)	-	<0.001
Inhaler Therapy					<0.01
None	96 (41.2)	16 (15.4)	24 (45.3)	56 (73.7)	
LABA	3 (1.3)	2 (1.9)	1 (1.9)	0 (0)	
LAMA	9 (3.8)	6 (5.8)	2 (3.8)	1 (1.3)	
ICS	15 (6.4)	7 (6.7)	3 (5.6)	5 (6.6)	
LABA/LAMA	13 (5.5)	13 (12.5)	0 (0)	0 (0)	
ICS/LABA	42 (18.1)	16 (15.4)	13 (24.5)	13 (17.1)	
ICS/LABA/LAMA	55 (23.7)	44 (42.3)	10 (18.9)	1 (1.3)	

Comparison of the patient groups. Values are presented in Number (%) or Mean ± Standard deviation. CKD = Chronic kidney disease; FEV1 = Forced expiratory volume in 1 second; FVC = Forced vital capacity; LLN = Lower limit of normal; LABA = Long-acting beta agonist; LAMA = Long-acting muscarinic antagonist; ICS = Inhaled corticosteroids. * Other includes internal medicine specialists or general physician. # Calculated using the Global Lung Initiative 2012 online calculator.

**Table 2 medicina-59-00632-t002:** Admission course and treatment.

	N (%) or Mean ± SD				
	Total (N = 233)	Confirmed COPD (N = 104)	No COPD (N = 53)	Never had Spirometry (N = 76)	*p*-Value
Antibiotics	183	88	39	56	0.13
	(78.5)	(84.6)	(73.6)	(73.7)	
Bronchodilators (Short acting)	208	94	47	67	0.89
	(89.3)	(90.4)	(88.7)	(88.2)	
Systemic steroids	186	86	42	58	0.57
	(79.8)	(82.7)	(79.2)	(76.3)	
Heart failure therapy	39 (16.7)	16 (15.4)	8 (15.1)	15 (19.7)	0.25
Respiratory support	45 (19.4)	27 (26)	6 (11.5)	12 (15.8)	0.06
Pulmonologist consultation during admission	42	26	9	7	0.02
	(18.0)	(25.0)	(17.0)	(9.2)	
Pulmonologist consultation following admission	118	81	29	8	<0.01
	(50.6)	(77.9)	(54.7)	(10.5)	
COPD diagnosis remained after admission	229	104	51	74	0.17
	(98.3)	(100.0)	(96.2)	(97.4)	
Laboratory results					
WBC	11.4 ± 7	12.3 ± 5.6	10.6 ± 11	10.7 ± 4.6	<0.01
CRP	6.1 ± 9	6.5 ± 9	4.8 ± 7.8	6.4 ± 10.3	0.75
PCO2	49.7 ± 15.2	53 ± 16	44.5 ± 13.8	48.4 ± 14	<0.01

Admission course and treatment in the patient groups. F/U = follow up; Respiratory support = invasive or non-invasive mechanical ventilation; WBC = White blood cell count; PCO2 = partial pressure of carbon dioxide; CRP = C-reactive protein; *p* < 0.0045 considered to be significant.

## Data Availability

No new data were created or analyzed in this study. Data sharing is not applicable to this article.

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
