# Peer review of "How Accurate Is the Diagnosis of “Chronic Obstructive Pulmonary Disease” in Patients Hospitalized with an Acute Exacerbation?"

_medicina, 2023, doi:10.3390/medicina59030632_

Round 1

Reviewer 1 Report

The manuscript entitled “How Accurate Is the Diagnosis of "Chronic Obstructive Pulmonary Disease in Patients Hospitalized with an Acute Exacerbation” presented a retrospective study with cross-sectional analysis of patient subgroups admitted for COPD treatment, with pre-admission spirometry and confirmed diagnosis of COPD according to the GOLD criteria, without COPD and without spirometry. They found a high association between recurrent admissions for COPD and confirmed COPD diagnosis, but also a high proportion of patients without confirmed COPD according to the current GOLD criteria or available spirometry measures. The paper is well written and structured, but there are some things that need to be improved in order to be accepted for publication.

Main concerns:

Methods:

- The statistical tests used for data analysis compare more than two groups, which can indicate whether there is a significant difference between at least two of these groups, but do not provide the specific paired results.  It would be more meaningful to perform the test among all paired group combinations.

- In the methods section, are the groups of patients with no COPD and no spirometry measures pooled together for the logistic regression analysis? As this analysis uses a binary outcome, I assume that only two groups were involved. How was the univariate analysis performed, also pooling these two groups?

Results:

- The information in the first paragraph of the results section regarding the size of the different subgroups is repeated on page 6 (but in % values), subsection "Comparison of patient groups based on spirometry". It would be better to include the % values in this paragraph.

- Subsection "Patient characteristics": The information described in this text is already summarised in Table 1. In any case, the authors may wish to complement it rather than simply repeat the values.

- Remove (Table 1) from "Differences between patient groups" and add a reference to this table in the text. The same should be done for Table 2 in the "Admission course" subsection.

- Comparison of Patient Groups Based on Spirometry. This subsection can be deleted as the information is already given at the beginning of the results.

Minor comments.

-       replace & by and in “compared to the other groups (67.3% vs 47.2% & 53.9%) (p = 0.03).

-       during their first admission ... and following admission(s)

 -       Laboratory results: (mean = 12.34) the unit measure is missing

Reviewer 2 Report

The authors provide a case series of patients hospitalized with a primary diagnosis of COPD, finding that more than half did not have spirometric obstruction.

Major Concerns:

Those without spirometry did not have "documented" spirometry ( they may have had it at some point but it was not in the record).  Based on the other data presented I suspect most of these would be obstructed.  In any case, a different metric would be the % of those with spirometry who had obstruction- this would be nearly 70% and could be commented on.

Those with spirometry that was not obstructed had a mean FEV1 of < 50%,  suggestion  more than half of these were restricted ( which s part of the new COPDGene definition of COPD).

Round 2

Reviewer 2 Report

The authors have been responsive to comments